# The Hateful Memes Challenge:
# Detecting Hate Speech in Multimodal Memes

**Douwe Kiela**,* **Hamed Firooz**,* **Aravind Mohan,**

**Vedanuj Goswami, Amanpreet Singh, Pratik Ringshia, Davide Testuggine**

**Facebook AI**
{dkiela,mhfirooz,aramohan,vedanuj,asg,tikir,davidet}@fb.com

## Abstract

This work proposes a new challenge set for multimodal classification, focusing on detecting hate speech in multimodal memes. It is constructed such that unimodal models struggle and only multimodal models can succeed: difficult examples ("benign confounders") are added to the dataset to make it hard to rely on unimodal signals. The task requires subtle reasoning, yet is straightforward to evaluate as a binary classification problem. We provide baseline performance numbers for unimodal models, as well as for multimodal models with various degrees of sophistication. We find that state-of-the-art methods perform poorly compared to humans, illustrating the difficulty of the task and highlighting the challenge that this important problem poses to the community.

## 1  Introduction

In the past few years there has been a surge of interest in multimodal problems, from image captioning [8, 43, 88, 28, 68] to visual question answering (VQA) [3, 26, 27, 33, 71] and beyond. Multimodal tasks are interesting because many real-world problems are multimodal in nature—from how humans perceive and understand the world around them, to handling data that occurs "in the wild" on the Internet. Many practitioners believe that multimodality holds the key to problems as varied as natural language understanding [5, 41], computer vision evaluation [23], and embodied AI [64].

Their success notwithstanding, it is not always clear to what extent truly multimodal reasoning and understanding is required for solving many current tasks and datasets. For example, it has been pointed out that language can inadvertently impose strong priors that result in seemingly impressive performance, without any understanding of the visual content in the underlying model [15]. Similar problems were found in VQA [3], where simple baselines without sophisticated multimodal understanding performed remarkably well [94, 35, 1, 26] and in multimodal machine translation [18, 74], where the image was found to matter relatively little [13, 17, 7]. In this work, we present a challenge set designed to measure truly multimodal understanding and reasoning, with straightforward evaluation metrics and a direct real world use case.

Specifically, we focus on hate speech detection in multimodal memes. Memes pose an interesting multimodal fusion problem: Consider a sentence like "love the way you smell today" or "look how many people love you". Unimodally, these sentences are harmless, but combine them with an equally harmless image of a skunk or a tumbleweed, and suddenly they become mean. See Figure 1 for an

---

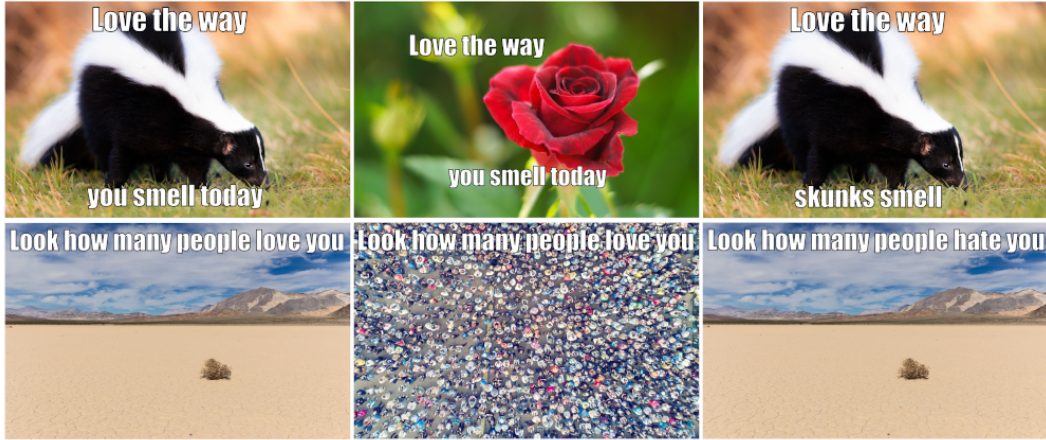

Figure 1: Multimodal "mean" memes and benign confounders, for illustrative purposes (not actually in the dataset; featuring real hate speech examples prominently in this place would be distasteful). Mean memes (left), benign image confounders (middle) and benign text confounders (right).

illustration. That is to say, memes are often subtle and while their true underlying meaning may be easy for humans to detect, they can be very challenging for AI systems.

A crucial characteristic of the challenge is that we include so-called "benign confounders" to counter the possibility of models exploiting unimodal priors: for every hateful meme, we find alternative images or captions that make the label flip to not-hateful. Using the example above, for example, if we replaced the skunk and tumbleweed images with pictures of roses or people, the memes become harmless again. Similarly, we can flip the label by keeping the original images but changing the text to "look how many people hate you" or "skunks have a very particular smell". This strategy is compatible with recent work arguing that tasks should include counterfactual or contrastive examples [38, 22, 26]. At the same time, the existence of such confounders is a good indicator of the need for multimodality—if both kinds (image and text confounders) exist, classifying the original meme and its confounders correctly will require multimodal reasoning. Thus, the challenge is designed such that it should only be solvable by models that are successful at sophisticated multimodal reasoning and understanding.

At the scale of the internet, malicious content cannot be tackled by having humans inspect every data point. Consequently, machine learning and artificial intelligence play an ever more important role in mitigating important societal problems, such as the prevalence of hate speech. Our challenge set can thus be said to serve the dual purpose of measuring progress on multimodal understanding and reasoning, while at the same time facilitating progress in a real-world application of hate speech detection. This further sets apart our challenge from existing tasks, many of which have limited or indirect real-world use cases.

In what follows, we describe in detail how the challenge set was created, analyze its properties and provide baseline performance numbers for various unimodal and state-of-the-art multimodal methods. We find that the performance of our baselines reflects a concrete hierarchy in their multimodal sophistication, highlighting the task's usefulness for evaluating the capabilities of advanced multimodal models. The gap between the best-performing model and humans remains large, suggesting that the hateful memes challenge is well-suited for measuring progress in multimodal research.

## 2 The Hateful Memes Challenge Set

The Hateful Memes dataset is a so-called challenge set, by which we mean that its purpose is not to train models from scratch, but rather to finetune and test large scale multimodal models that were pre-trained, for instance, via self-supervised learning. In order to reduce visual bias and to make sure we could obtain appropriate licenses for the underlying content, we reconstruct source memes from scratch using a custom-built tool. We used third-party annotators, who were trained on employing the hate speech definition used in this paper. Annotators spent an average time of 27 minutes per final

meme in the dataset. In this section, we describe how we define hate speech, how we obtain and annotate memes, and give further details on how the challenge set was constructed.

## 2.1 Hatefulness Definition

Previous work has employed a wide variety of definitions and terminology around hate speech. Hate speech, in the context of this paper and the challenge set, is strictly defined as follows:

> A direct or indirect attack on people based on characteristics, including ethnicity, race, nationality, immigration status, religion, caste, sex, gender identity, sexual orientation, and disability or disease. We define attack as violent or dehumanizing (comparing people to non-human things, e.g. animals) speech, statements of inferiority, and calls for exclusion or segregation. Mocking hate crime is also considered hate speech.

This definition resembles community standards on hate speech employed by e.g. Facebook[2]. There has been previous work arguing for a three-way distinction between e.g. hateful, offensive and normal [11], but this arguably makes the boundary between the first two classes murky, and the classification decision less actionable.

The definition employed here has some notable exceptions, i.e., attacking individuals/famous people is allowed if the attack is not based on any of the protected characteristics listed in the definition. Attacking groups perpetrating hate (e.g. terrorist groups) is also not considered hate. This means that hate speech detection also involves possibly subtle world knowledge.

## 2.2 Reconstructing Memes using Getty Images

One important issue with respect to dataset creation is having clarity around licensing of the underlying content. We've constructed our dataset specifically with this in mind. Instead of trying to release original memes with unknown creators, we use "in the wild" memes to manually reconstruct new memes by placing, without loss of meaning, meme text over a new underlying stock image. These new underlying images were obtained in partnership with Getty Images under a license negotiated to allow redistribution for research purposes. This approach has several additional benefits: 1) it allows us to avoid any potential noise from optical character recognition (OCR), since our reconstruction tool records the annotated text; and 2) it reduces any bias that might exist in the visual modality as a result of e.g. using different meme creation tools or underlying image sources.

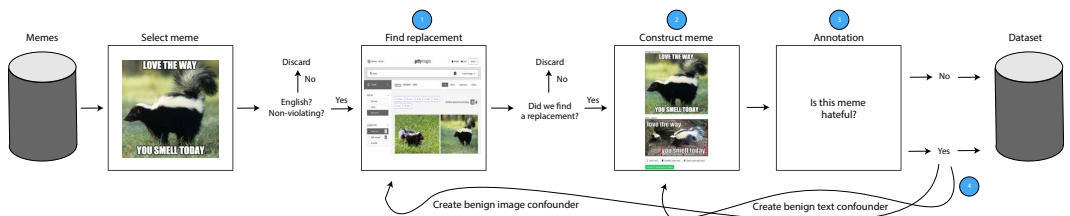

Figure 2: Flowchart of the annotation process

## 2.3 Annotation Process

We used a third-party annotation company rather than a crowdsourcing platform. The annotators were trained for 4 hours in recognizing hate speech according to the above definition. Three pilot runs were subsequently performed where annotators were tasked with classifying memes and were then given feedback to improve their performance.

The annotation process comprised several steps, which are shown in Figure 2. To summarize: for a set of memes occurring in the wild, which we use as our source set, we 1) find replacement images that we can license from Getty; 2) use these replacement images to construct novel memes using the original text; and 3) subsequently have these annotated for hatefulness (by a different set of

annotators). For memes found to be hateful, we also 4) construct benign confounders, if possible, by replacing the image and/or the text of the newly created hateful meme with an alternative that flips the label from hateful back to not-hateful. We describe each step in detail in the following sections.

### 2.3.1 Phase 1: Filtering

The annotation process started from a source dataset of more than 1 million images occurring in the wild, posted on public social media groups from within the United States. The content was filtered to discard non-meme images and deduplicated, yielding a set of 162k memes, which served as the source set from which we construct our set of newly created memes.

In Phase 1, we asked annotators to check that the meme text was in English, and that the meme was non-violating. Memes were considered violating if they contained any of the following: self injury or suicidal content, child exploitation or nudity, calls to violence, adult sexual content or nudity, invitation to acts of terrorism and human trafficking. Lastly, we removed content that contained slurs since these are unimodal by definition. This resulted in a set of roughly 46k candidate memes. For the memes that passed this test, we asked annotators to use the Getty Images search API to find similar images to the meme image, according the following definition:

> an image (or set of images) is a suitable replacement if and only if, when overlaid with the meme's text, it *preserves the meaning and intent of the original meme*.

In other words, images only constitute suitable replacements if they preserve the semantic content of the original meme that was used for inspiration. If no meaning-preserving replacement image was found, the meme was discarded.

### 2.3.2 Phase 2: Meme construction

The process from Phase 1 yielded a set of image replacement candidates for the original set of non-discarded inspiration memes. In the next phase, we asked annotators to use a custom-designed meme creation tool to create a novel meme using the new underlying image (see Figure 3 for an example of the interface). Text from the original meme was extracted using an off-the-shelf OCR system and used to provide the annotator with a starting point for construction.

The annotator was shown the original meme and asked to reproduce it as faithfully as possible using the new underlying image, focusing on the preservation of semantic content. Some memes consist of multiple background images, in which case these were first vertically stacked and provided as a single image to the user. The annotators stored the resultant meme in PNG and SVG formats (the latter gives flexibility for potentially moving or augmenting the text, as well as extracting it or removing it altogether).

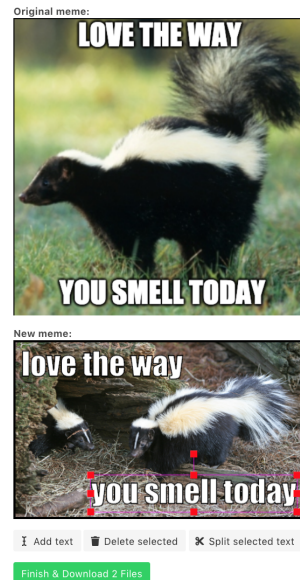

Figure 3: Custom interface for meme reconstruction.

### 2.3.3 Phase 3: Hatefulness rating

Next, the annotators were presented with the newly created memes, and asked to provide a hatefulness rating according to the above definition. Each meme was rated by three different annotators on a scale of 1-3, with a 1 indicating definitely hateful; a 2 indicating not sure; and a 3 indicating definitely not-hateful. For every meme we collected five annotations. Ternary rather than binary labelling was useful for filtering memes with severe disagreement (e.g. two annotators voting 1 and three voting 3), which we had annotated by an expert. This yielded a single binary classification label for each of the newly constructed memes.

### 2.3.4 Phase 4: Benign confounders

Finally, for the subset of memes that were annotated as hateful, we collected confounders (a.k.a., "contrastive" [22] or "counterfactual" [38] examples). This allows us to address some of the biases that machine learning systems would easily pick up on. For instance, a system might pick up on

accidental biases where appearances of words like "black" or "white" might be strongly correlated with hate speech. To mitigate this problem, and to make the dataset extra challenging, we collect benign confounders, defined as follows:

> A benign confounder is a minimum replacement image or replacement text that flips the label for a given multimodal meme from hateful to non-hateful.

For example, in the "love the way you smell today" example, we could replace the image of the skunk with an image of roses, or conversely we could replace the text with "skunks have a very particular smell". In both cases, the meme would no longer be mean, but just a statement—as such, the label would be flipped. These benign confounders make the dataset more challenging, and make it so that any system, if it wants to solve this task, *has to be multimodal*: using just the text, or just the image, is going to lead to poor performance.

The benign confounder construction phase also allows us to spot memes that are only unimodally hateful: if it is not possible to find a replacement text that flips the label, the image itself is hateful. The same is true in the other direction (if no replacement image can flip the label), but somewhat more complicated, since it might be that it was simply not possible to find a suitable replacement image. In our experiments, we found that almost all cases of unimodal hate were text-only, so this is an important category of memes.

## 2.4 Splits

Using the various phases outlined above and after further filtering to remove low-quality examples, we end up with a dataset totalling exactly 10k memes. The dataset comprises five different types of memes: *multimodal hate*, where benign confounders were found for both modalities, *unimodal hate* where one or both modalities were already hateful on their own, *benign image* and *benign text* confounders and finally *random not-hateful* examples.

We construct a dev and test set from 5% and 10% of the data respectively, and set aside the rest to serve as fine-tuning training data. The dev and test set are fully balanced, and are comprised of memes using the following percentages: 40% multimodal hate, 10% unimodal hate, 20% benign text confounder, 20% benign image confounder, 10% random non-hateful.

The objective of the task is, given an image and the pre-extracted—i.e., we do not require OCR—text, to classify memes according to their hatefulness. We recommend that people report the area under the receiver operating characteristic curve (ROC AUC) [6] as the main metric, but also report accuracy since the test set is balanced and that metric is easier to interpret.

## 2.5 Competition and "unseen" test set

A NeurIPS competition will be held based on the dataset described in this paper. The winners will be determined according to performance on a different "unseen" test set. For more information about the competition, please visit https://ai.facebook.com/hatefulmemes.

## 3 Dataset Analysis

In this section, we analyze the properties of the dataset in more detail, and examine inter annotator agreement and the presence of different types of hate category and type of attack found in our (not necessarily representative) data sample.

## 3.1 Inter-Annotator Agreement

As explained in Section 2.3.3, memes were reviewed by three annotators. We use Cohen's kappa score to measure the inter annotators reliability in hatefulness rating. We found the score to be 68.4 which indicates "moderate" agreement [55]. The score demonstrates the difficulty of determining whether content constitutes hate speech or not. Furthermore, the hate speech definition can be rather complex and contains exceptions. For example, attacking a terrorist organization is not hateful but equating a terrorist organization with people of some race or ethnicity is considered hateful.

| Type | Model | Validation | | Test | |
|------|-------|:----:|:----:|:----:|:----:|
| | | Acc. | AUROC | Acc. | AUROC |
| | Human | - | - | 84.70 | - |
| Unimodal | Image-Grid | 50.67 | 52.33 | 52.73±0.72 | 53.71±2.04 |
| | Image-Region | 52.53 | 57.24 | 52.36±0.23 | 57.74±0.73 |
| | Text BERT | 58.27 | 65.05 | 62.80±1.42 | 69.00±0.11 |
| Multimodal (Unimodal Pretraining) | Late Fusion | 59.39 | 65.07 | 63.20±1.09 | 69.30±0.33 |
| | Concat BERT | 59.32 | 65.88 | 61.53±0.96 | 67.77±0.87 |
| | MMBT-Grid | 59.59 | 66.73 | 62.83±2.04 | 69.49±0.59 |
| | MMBT-Region | 64.75 | 72.62 | 67.66±1.39 | 73.82±0.20 |
| | ViLBERT | 63.16 | 72.17 | 65.27±2.40 | 73.32±1.09 |
| | Visual BERT | 65.01 | 74.14 | 66.67±1.68 | 74.42±1.34 |
| Multimodal (Multimodal Pretraining) | ViLBERT CC | 66.10 | 73.02 | 65.90±1.20 | 74.52±0.06 |
| | Visual BERT COCO | 65.93 | 74.14 | 69.47±2.06 | 75.44±1.86 |

Table 1: Model performance.

In cases with good agreement, we followed the majority vote for deciding the binary label. For cases where there was significant crowdworker disagreement, we had the meme checked and the correct label assigned by expert annotators. As noted, if even those disagreed, the meme was discarded. The dev set and test set were double checked by an expert in their entirety. We had a different set of annotators label the final test set, and obtained a human accuracy of 84.7% with low variance (ten batches were annotated by separate annotators, per-batch accuracy was always in the 83-87% range).

## 3.2 Hate Categories and Types of Attack

Appendix C shows an analysis of the dataset in terms of the categories of hate and the types of attack, shedding more light on how the dataset characteristics relate to the hate speech definition.

## 3.3 Lexical Statistics

Appendix B shows an analysis of the lexical statistics of the dataset, including of the visual modality.

## 4 Benchmarking Multimodal Classification

In this section, we establish baseline scores for various unimodal and several state-of-the-art multimodal models on this task. We find that performance relative to humans is still poor, indicating that there is a lot of room for improvement. Starter kit code is available at `https://github.com/facebookresearch/mmf/tree/master/projects/hateful_memes`.

## 4.1 Models

We evaluate a variety of models, belonging to one of the three following classes: unimodal models, multimodal models that were unimodally pretrained (where for example a pretrained BERT model and a pretrained ResNet model are combined in some way), and multimodal models that were multimodally pretrained (where a model was pretrained using a multimodal objective).

We evaluate two image encoders: 1) standard ResNet-152 [30] convolutional features from `res-5c` with average pooling (**Image-Grid**) 2) features from `fc6` layer of Faster-RCNN [60] with ResNeXt-152 as its backbone [86]. The Faster-RCNN is trained on Visual Genome [43] with attribute loss following [69] and features from `fc6` layer are fine-tuned using weights of the `fc7` layer (**Image-Region**). For the textual modality, the unimodal model is BERT [14] (**Text BERT**).

Transfer learning from pre-trained text and image models on multimodal tasks is well-established, and we evaluate a variety of models that fall in this category. We include results for simple fusion methods where we take the mean of the unimodal ResNet-152 and BERT output scores (**Late Fusion**) or concatenate ResNet-152 features with BERT and training an MLP on top (**Concat**

**BERT**). These can be compared to more sophisticated multimodal methods, such as supervised multimodal bitransformers [40] using either Image-Grid or Image-Region features (**MMBT-Grid** and **MMBT-Region**), and versions of ViLBERT [50] and Visual BERT [46] that were only unimodally pretrained and not pretrained on multimodal data (**ViLBERT** and **Visual BERT**).

Finally, we compare these methods to models that were trained on a multimodal objective as an intermediate step before we finetune on the current task. Specifically, we include the official multi-modally pretrained versions in the ViLBERT (trained on Conceptual Captions [66], **ViLBERT CC**) and Visual BERT (trained on COCO, **Visual BERT COCO**).

We performed grid search hyperparameter tuning over the learning rate, batch size, warm up and number of iterations. We report results averaged over three random seeds, together with their standard deviation. See Appendix A for more details.

### 4.2 Results

The results are shown in Table 1. We observe that the text-only classifier performs slightly better than the vision-only classifier. Since the test set is balanced, the random and majority-class baselines lie at 50 AUROC exactly, which gets only marginally outperformed by the visual models.

The multimodal models do better. We observe that the more advanced the fusion, the better the model performs, with early fusion models (MMBT, ViLBERT and Visual BERT) broadly outperforming middle (Concat) and late fusion approaches.

Interestingly, we can see that the difference between unimodally pretrained models and multimodally pretrained models pretraining is relatively small, corroborating findings recently reported in [70] and indicating that multimodal pretraining can probably be improved further.

Accuracy for trained (but non-expert) annotators is 84.7%. The fact that even the best multimodal models are still very far away from human performance shows that there is room for improvement.

## 5 Related Work

**Hate speech** There has been a lot of work in recent years on detecting hate speech in network science [61] and natural language processing [81, 65, 19]. Several text-only hate speech datasets have been released, mostly based on Twitter [83, 82, 11, 24, 20], and various architectures have been proposed for classifiers [45, 54, 53]. Hate speech detection has proven to be difficult, and for instance subject to unwanted bias [16, 63, 10]. One issue is that not all of these works have agreed on what defines hate speech, and different terminology has been used, ranging from offensive or abusive language, to online harassment or aggression, to cyberbullying, to harmful speech, to hate speech [81]. Here, we focus exclusively on hate speech in a narrowly defined context.

**Multimodal hate speech** There has been surprisingly little work related to multimodal hate speech, with only a few papers including both images and text. Yang et al. [87] report that augmenting text with image embedding information immediately boosts performance in hate speech detection. Hosseinmardi et al. [31] collect a dataset of Instagram images and their associated comments, which they then label with the help of Crowdflower workers. They asked workers two questions: 1) does the example constitute cyberaggression; and 2) does it constititute cyberbullying. Where the former is defined as "using digital media to intentionally harm another person" and the latter is a subset of cyber-aggression, defined as "intentionally aggressive behavior that is repeatedly carried out in an online context against a person who cannot easily defend him or herself" [31]. They show that including the image features improves classification performance. The dataset consisted of 998 examples, of which 90% was found to have high-confidence ratings, of which 52% was classified as bullying. Singh et al. [72] conduct a detailed study, using the same dataset, of the types of features that matter for cyber-bullying detection in this task. Similarly, Zhong et al. [93] collected a dataset of Instagram posts and comments, consisting of 3000 examples. They asked Mechanical Turk workers two questions: 1) do the comments include any bullying; and 2) if so, is the bullying due to the content of the image. 560 examples were found to be bullying. They experiment with different kinds of features and simple classifiers for automatically detecting whether something constitutes bullying.

Our work differs from these works in various ways: our dataset is larger and explicitly designed to be difficult for unimodal architectures; we only include examples with high-confidence ratings from trained annotators and carefully balance the dataset to include different kinds of multimodal fusion problems; we focus on hate speech, rather than the more loosely defined cyberbullying; and finally we test more sophisticated models on this problem. Vijayaraghavan et al. [78] propose methods for interpreting multimodal hatespeech detection models, where the modalities consist of text and socio-cultural information rather than images. Concurrently, Gomez et al. [25] introduced a larger (and arguably noisier) dataset for multimodal hate speech detection based on Twitter data, which also contains memes and which would probably be useful as pretraining data for our task.

**Vision and language tasks**   Multimodal hate speech detection is a vision and language task. Vision and language problems have gained a lot of traction is recent years (see Mogadala et al. [56] for a survey), with great progress on important problems such as visual question answering [3, 26] and image caption generation and retrieval [8, 88, 43, 68, 28], with offshoot tasks focusing specifically on visual reasoning [36], referring expressions [39], visual storytelling [57, 32], visual dialogue [9, 12], multimodal machine translation [18, 74], visual reasoning [75, 33, 71, 85, 27], visual common sense reasoning [91] and many others.

A large subset of these tasks focus on (autoregressive) text generation or retrieval objectives. One of the two modalities is usually dominant. They often rely on bounding boxes or similar features for maximum performance, and are not always easy to evaluate [77]. While these tasks are of great interest to the community, they are different from the kinds of real-world multimodal classification problems one might see in industry—a company like Facebook or Twitter, for example, needs to classify a lot of multimodal posts, ads, comments, etc for a wide variety of class labels. These use cases often involve large-scale, text-dominant multimodal classification similar to what is proposed in this task.

Related multimodal classification tasks exist; for instance, there has been extensive research in multimodal sentiment [73], but there is no agreed-upon standard dataset or benchmark task. Other datasets using internet data include Food101 [80], where the goal is to predict the dish of recipes and images; various versions of Yelp reviews [52]; Walmart and Ferramenta product classification [90, 21]; social media name tagging (Twitter and Snapchat) [49]; social media target-oriented sentiment [89]; social media crisis handling [2]; various multimodal news classification datasets [59, 67]; multimodal document intent in Instagram posts [44]; and predicting tags for Flickr images [76, 37]. Other datasets include grounded entailment, which exploits the fact that one of the large-scale natural language inference datasets was constructed using captions as premises, yielding a image, premise, hypothesis triplet with associated entailment label [79]; as well as MM-IMDB, where the aim is to predict genres from posters and plots [4]; and obtaining a deeper understanding of multimodal advertisements, which requires similarly subtle reasoning [34, 92]. Sabat et al. [62] recently found in a preliminary study that the visual modality can be more informative for detecting hate speech in memes than the text. The quality of these datasets varies substantially, and their data is not always readily available to different organizations. Consequently, there has been a practice where authors opt to simply "roll their own" dataset, leading to a fragmented status quo. We believe that our dataset fills up an important gap in the space of multimodal classification datasets.

# 6   Conclusion

In this work, we introduced a new challenge dataset and benchmark centered around detecting hate speech in multimodal memes. Hate speech is an important societal problem, and addressing it requires improvements in the capabilities of modern machine learning systems. Detecting hate speech in memes requires reasoning about subtle cues and the task was constructed such that unimodal models find it difficult, by including "benign confounders" that flip the label of a multimodal hateful meme. Our analysis provided insight into the distribution of protected categories and types of attack, as well as words and object frequencies. We found that results on the task reflected a concrete hierarchy in multimodal sophistication, with more advanced fusion models performing better. Still, current state-of-the-art multimodal models perform relatively poorly on this dataset, with a large gap to human performance, highlighting the challenge's promise as a benchmark to the community. We hope that this work can drive progress in multimodal reasoning and understanding, as well as help solve an important real-world problem.

## Broader Impact

This work offers several positive societal benefits. Hate speech is a well-known problem, and countering it via automatic methods can have a big impact on people's lives. This challenge is meant to spur innovation and encourage new developments in multimodal reasoning and understanding, which can have positive effects for an extremely wide variety of tasks and applications. With these advantages also come potential downsides: better multimodal systems may lead to the automation of jobs in the coming decades, and could be used for censorship or nefarious purposes. These risks can in part be mitigated by developing AI systems to counter them.

## Footnotes

[2]https://www.facebook.com/communitystandards/hate_speech

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
