[Supplementary Material]

# A  Implementation Details & Hyperparameters

|  | | HyperParam | | | | |
|---|---|---|---|---|---|---|
| Model | Batch Size | Peak LR | Total Parameters | Classifier Layers | Text Encoder | Image Encoder |
| Image-Grid | 32 | 1e-5 | 60312642 | 2 | N/A | ResNet-152 |
| Image-Region | 64 | 5e-5 | 6365186 | 1 | N/A | FasterRCNN |
| Text BERT | 128 | 5e-5 | 110668034 | 2 | BERT | N/A |
| Late Fusion | 64 | 5e-5 | 170980676 | 2 | BERT | ResNet-152 |
| Concat BERT | 256 | 1e-5 | 170384706 | 2 | BERT | ResNet-152 |
| MMBT-Grid | 32 | 1e-5 | 169793346 | 1 | BERT | ResNet-152 |
| MMBT-Region | 32 | 5e-5 | 115845890 | 1 | BERT | FasterRCNN |
| ViLBERT | 32 | 1e-5 | 247780354 | 2 | BERT | FasterRCNN |
| Visual BERT | 128 | 5e-5 | 112044290 | 2 | BERT | FasterRCNN |
| ViLBERT CC | 32 | 1e-5 | 247780354 | 2 | BERT | FasterRCNN |
| Visual BERT COCO | 64 | 5e-5 | 112044290 | 2 | BERT | FasterRCNN |

Table 2: Hyperparameters used for different models.

In Table 2, we list out the different hyperparameter settings for our experiments. The exact reproduction details and scripts for the experiments are available with code. Grid search was performed on batch size, learning rate and maximum number of updates to find the best hyperparameter configuration. While batch size and learning rate differ based on the model as shown in Table A, we found 22000 updates as best performing. The models were evaluated at an interval of 500 updates on the dev set and the model with best AUROC was taken as the final model to be evaluated on test set. We use weighted Adam [42, 48] with cosine learning rate schedule and fixed 2000 warmup steps [51] for optimization without gradient clipping. The $\epsilon$, $\beta_1$ and $\beta_2$ for Adam is set to 1e-8, 0.9 and 0.98 respectively. The models were implemented in PyTorch [58] using MMF [69]. We trained all of the models on 2 nodes each containing 8 V100 GPUs in a distributed setup.

For ViLBERT, we use the default setting of 6 and 12 TRM blocks for the visual and linguistic streams respectively and use the original weights pretrained on Conceptual Captions [66] provided with [50]. For VisualBERT, we follow [46] and use 12 TRM blocks and use COCO [47] pretrained weights provided with [70]. The hidden size for textual stream TRM blocks is set to 768 and for visual stream is set to 1024. The dropout value for linear and attention layers is set to 0.1 in TRM blocks. The dropout for classifier MLP layers is set to 0.5. For BERT text encoder, we use pretrained BERT base uncased weights provided in the Transformers library [84]. For region features from Faster RCNN, we use features from fc6 layer and fine-tune the weights for fc7 layer. We fine-tune all models fully end-to-end without freezing anything regardless of whether model was pretrained or not.

# B  Textual and Visual Lexical Statistics

Figure 4: Histograms, normalized, of text length by type (left) and by class (right).

In this section, we analyze the lexical (word-level) statistics of the dataset. Figure 4 shows the token count distribution, broken down by binary hatefulness label, as well as by category (multimodal hate, unimodal hate, etc). We also analyze the frequency of words for different classes in the test set. Table 3 shows the top 10 most-frequent words by class and their normalized frequency. We observe that certain words used in dehumanizing based on gender (e.g. equating women with "dishwashers"

| mm hate | um hate | text benign | image benign | not-hateful |
|---|---|---|---|---|
| like (0.05) | people (0.14) | like (0.05) | like (0.06) | want (0.08) |
| white (0.04) | like (0.07) | love (0.05) | dishwasher (0.05) | think (0.05) |
| people (0.04) | get (0.07) | people (0.05) | one (0.05) | get (0.05) |
| black (0.04) | i'm (0.05) | day (0.04) | get (0.04) | know (0.05) |
| get (0.04) | muslims (0.05) | time (0.04) | i'm (0.04) | people (0.05) |
| one (0.04) | black (0.05) | one (0.04) | way (0.03) | like (0.05) |
| dishwasher (0.03) | white (0.05) | take (0.03) | white (0.03) | take (0.04) |
| i'm (0.03) | us (0.03) | world (0.03) | islam (0.03) | always (0.04) |
| know (0.03) | america (0.03) | man (0.02) | black (0.03) | say (0.04) |
| back (0.02) | go (0.03) | look (0.02) | gas (0.03) | trump (0.04) |

Table 3: Textual lexical analysis: Most frequent non-stopwords in the combined dev and test sets.

| mm hate | um hate | text benign | image benign | not-hateful |
|---|---|---|---|---|
| person (3.43) | person (2.27) | person (3.33) | person (2.13) | person (2.11) |
| tie (0.16) | tie (0.21) | tie (0.18) | bird (0.16) | tie (0.23) |
| car (0.10) | cell phone (0.08) | dog (0.12) | chair (0.12) | cup (0.12) |
| chair (0.10) | dog (0.07) | book (0.12) | book (0.11) | chair (0.09) |
| book (0.07) | car (0.07) | car (0.11) | car (0.09) | car (0.07) |
| dog (0.07) | bird (0.06) | chair (0.08) | cup (0.08) | cell phone (0.06) |
| cell phone (0.05) | chair (0.05) | sheep (0.05) | dog (0.08) | dog (0.05) |
| handbag (0.05) | tennis racket (0.05) | cell phone (0.05) | bowl (0.08) | bottle (0.03) |
| sheep (0.04) | cat (0.03) | bottle (0.04) | sheep (0.08) | bed (0.03) |
| bottle (0.04) | cup (0.03) | knife (0.04) | bottle (0.07) | teddy bear (0.03) |

Table 4: Visual lexical analysis: Most frequent Mask-RCNN labels in the combined dev and test sets.

or "sandwich makers") and colors ("black" and "white") are frequent. Note that these are also frequent in the benign image confounder category, meaning that these words are not necessarily directly predictive of the label. In unimodal hate (which is almost always text-only), we observe that the language is stronger and often targets religious groups.

Shedding light on the properties of the visual modality, rather than the text, is more difficult. We do the same analysis but this time for bounding box labels from Mask R-CNN [29]. The results are in Table 4. Aside from the fact that these systems appear quite biased, we find that the dataset seems relatively evenly balanced across different properties and objects.

## C Hate Categories and Types of Attack

### C.1 Hate Categories

According to the definition in Section 2.1, hate speech is defined using the characteristics of certain protected categories. That is, creators and distributors of hateful memes aim to incite hatred, enmity or humiliation of groups of people based on the grounds of gender, nationality, etc. and if that group falls in one of the protected categories, it constitutes hate speech under our definition. We analyzed the dev set based on these protected categories. In our analysis, we identified nine such protected categories as listed in Table 5 (right). One meme can attack multiple protected categories. We observe that race and religion-based hate are the most prevalent. Note that this set of protected categories is by no means complete or exhaustive, but it is useful to get a better sense of the dataset properties.

### C.2 Types of Attack

Hateful memes employ different types of attacks for different protected categories. In our analysis, we classified these attack types into different classes, together with a catch-all "other" class. One of the major types of attack is dehumanization, in which groups of people are compared with non-human things, such as animals or objects. We break this down into different types of dehumanization found

| Hate speech type | % |
|---|---|
| Comparison to animal | 4.0 |
| Comparison to object | 9.2 |
| Comparison w criminals | 17.2 |
| Exclusion | 4.0 |
| Expressing Disgust/Contempt | 6.8 |
| Mental/physical inferiority | 7.2 |
| Mocking disability | 6.0 |
| Mocking hate crime | 14.0 |
| Negative stereotypes | 15.6 |
| Other | 4.4 |
| Use of slur | 2.0 |
| Violent speech | 9.6 |

| Protected category | % |
|---|---|
| Race or Ethnicity | 47.1 |
| Religion | 39.3 |
| Sexual Orientation | 4.9 |
| Gender | 14.8 |
| Gender Identity | 4.1 |
| Disability or Disease | 8.2 |
| Nationality | 9.8 |
| Immigration Status | 6.1 |
| Socioeconomic Class | 0.4 |

Table 5: Annotation by hate speech type and protected category of the dev set. Multiple labels can apply per meme so percentages do not sum to 100.

in the data. Attack types are exclusive (only one per meme) and their frequencies are listed in Table 5 (left). Our findings show that the types of attack are broadly distributed, with comparison with criminals (e.g. terrorists), negative stereotypes (e.g. saying that certain demographics are sexually attracted to children/animals) and mocking hate crimes (e.g. the holocaust) being the most frequent categories. Again, note that this set of attack types is by no means complete or exhaustive, but it helps us understand what types of attack a classifier needs to detect if it is to be successful on the task.