[Reviews · NeurIPS 2020]

Review 1

Summary and Contributions: The paper proposes a challenge in detecting hate speech in multimodal memes (text + image; the problem is posed as binary classification). The dataset is constructed in a way such that it requires models to perform "multimodal reasoning" to succeed. The authors perform baseline evaluation using a set of uni- and multimodal models and show the performance of these models is inferior to human performance by a considerable margin.

Strengths: - An interesting task with both research (multimodal reasoning) and practical (social media moderation) applications. - Careful construction of the challenge dataset that makes it difficult for systems to "cheat" by exploiting solely a single modality. - Detailed framing of the work in related literature. - There are plans to run the challenge as a public competition with an "unseen" test set. - An interesting result indicating that there is room to grow in terms of multimodal pretraining, as the difference between unimodally and multimodally pretrained models is relatively small (some unimodally pretrained models perform better than the multimodal ViLBERT CC model).

Weaknesses: - Main weakness: the results section contains very little analysis. At the very least, it'd be useful to indicate how accuracy differs across different classes of memes in the test dataset (multimodal vs unimodal hate, benign image/text, other random non-hateful). - Additional analyses could be performed on the dev set, using the annotations from the appendix Table B.1. - The dataset is not large (10k items) and it is therefore unclear whether considerable gains in performance can be attained by constructing a few thousand additional memes. An additional evaluation (at least of the top performing model/s) using subsets of the training set of different sizes could shed some light on this. - Generalization to real world memes may be limited by the fact that a single tool was used to generate all the memes in the dataset (discussion in section 2.2).

Correctness: Yes.

Clarity: Yes.

Relation to Prior Work: Yes.

Reproducibility: Yes

Additional Feedback: - It'd be helpful to indicate in Table 1 which rows correspond to early/middle/late fusion to ease reading the results. - Line 236 is missing a closing parenthesis. - It's not clear how exactly the unimodal versions of VilBERT and Visual BERT were trained. - Should the appendix Table B.1 be labelled Table C.1 (as it relates to Appendix C)? - Line 175: "further filtering to remove low-quality examples" - can you detail what kind of filtering you performed here? --after author response-- - Additional analyses described in the response (varying training dataset sizes, model failure modes) will make the paper and the remaining challenges clearer.


Review 2

Summary and Contributions: The paper introduces a novel dataset (Hates Memes Challenge) with 10k annotated image+text memes. In the paper, authors detail the procedures of the data collection and annotation. The memes in the dataset contain the associated text in text form as well as the proper licence to be used. Finally, authors evaluate many unimodal and multi-modal models on their dataset, and show how multimodality is relevant to correctly solve the task.

Strengths: - The dataset contains counterfactual examples, which enable machine learning practitioners to improve their models through better language-image modelling. - The dataset is well curated, the meme annotation seem to be rigorous and the final labels are high quality. - The topic detecting hateful content in internet is very relevant on our times. Improving systems to detect hateful speech is going to improve the quality of the public debate as well as reduce attacks over internet. I think the topic is relevant and the contribution significant. - Model performance reported by the authors seem to indicate that actual models are still far from human performance. This means there is still a lot of work to do in the topic, and the dataset will provide the community a good reference to advance in the topic.

Weaknesses: - In my opinion, the main weakness of the paper is the reduced size of the dataset. Authors propose a dataset with around 10k examples, from which 10% of it is used for testing. This means, the test performance which will be driving the field and the performance in the task will be computed over 1000 samples, which in my opinion is very limited. From the paper description, I understand the annotation is very costly, but I wonder if the fact that the labels are clean compensates the small amount of overall data. I believe authors should analyse in depth the effect of dataset size in the models. - Did the authors check whether there is an added bias when using images from Getty? As the images do not come from the original memes, it could be that the distribution is different between the newly defined memes and the original ones. - It would be interesting to look at the failure modes of the models. Are the models consistent on their failures? Are there particularly hard categories in the task? - How are the authors dealing with non-standard text (acronyms, non existent words, etc)? As memes are collected in the wild, I would assume some of the text is non standard. Are authors correcting the text if that happens?

Correctness: Yes. The authors describe the methodology very well in the paper and all the claims and methodology looks correct.

Clarity: Yes, the paper is very clear and well written. Authors clearly describe the different steps to build the dataset as well as the baselines used.

Relation to Prior Work: Yes, the authors highlight in the related work the difference between previously published works on hate speech and their dataset and approach.

Reproducibility: Yes

Additional Feedback: - L236 there is a typo with a parenthesis not closing.


Review 3

Summary and Contributions: This paper proposes a new dataset for multimodal hate speech detection. This dataset is carefully designed to be difficult for unimodal prediction. Several baseline models including unimodal and multimodal models are provided in the experiment part. This paper also finds that state-of-the-art methods perform poorly compared to humans.

Strengths: + This paper collects a new dataset especially for multimodal hate speech detection and the dataset may be beneficial to the relevant research community. + This paper exhibits the detailed annotation process elaborately, which provides a new method for another dataset collection. + Both unimodal and multimodal baseline models are discussed in the paper.

Weaknesses: - This paper is not well-organized. A proper dataset paper should emphasize the dataset analysis and comparison with the previous datasets. However, this paper turns to focus on the tedious annotation process, which is not that important and ought to be put in the supplementary materials. By and large, this paper is more like a technical report, not a standard and qualified NeurIPS paper. - The detailed dataset analysis can't be found in the paper. How large is the proposed dataset? What's the superiority over the previous datasets? As illustrated in the definition of hatefulness in part 2.1, the hate speech should be divided into several categories naturally. Therefore, the proposed dataset should be highly structured. How about the structure of the dataset? For example, the authors can provide several samples of ethnicity and religion. - As mentioned in the introduction part, the determination of multimodal memes is often subtle. Different people may have different opinions for the same memes. Must the opposite of the hateful speech be harmless ones? Hate speech detection is NOT a strictly binary classification problem. Hence, the modeling for the task in the paper is inaccurate. Could the authors present a candidate solution for it? - The baseline models provided in the experiment part is too simple. More advanced multimodal information fusion methods, like gated-fusion, should be further explored. Gated-fusion paper: DeepDualMapper: A Gated Fusion Network for Automatic Map Extraction using Aerial Images and Trajectories. AAAI 2020.

Correctness: The claims are correct. But the method exists a problem. The authors should present a structured dataset, not a random-organized one.

Clarity: No, the structure of the paper exists a big problem referred in the weakness part.

Relation to Prior Work: No. The comprehensive comparison with the datasets in [25] and [78] is NOT presented at all. Actually, only a clear comparison with the previous dataset can make the proposed dataset meaningful and stand out.

Reproducibility: Yes

Additional Feedback: ====== Post Rebuttal ======= More analysis and comparison with previous datasets should be provided. I decide to give 5 finally.


Review 4

Summary and Contributions: This paper introduces a newly created dataset, Hateful Memes. This dataset is intended for evaluating multimodal understanding models on the hateful memes detection task, in which the model takes a meme (image + text) and predicts if it's hateful or not. The dataset is created by experienced and trained annotators from a thrid-party company, which ensures the data quality. Experiments on a large variety of baseline models showed that even the best model largely underperforms human, implying that the dataset is challenging.

Strengths: 1. The hateful memes detection task is of great importance in practice. For example, as mentioned in paper, it can help with controlling malicious contents on social media. 2. The dataset was created carefully. The concept of "hateful" is rigorously defined and followed in the dataset collection process. Annotators are from a third-party company instead of crowdsourcing platforms. Each annotator were trained for 4 hours with feedbacks to improve their performance. When annotators disagree, there are expert annotators to make further decisions. Also, the dataset is reasonably large (10K examples), given the rigorousness. 3. Adding the "benign confounders" makes the dataset not easily solvable by unimodal models, thus requiring "real" multimodal understanding. This makes the dataset potentially very helpful for the multimodality area, since (as mentioned in paper) for many current benchmarks, complex unimodal models already achieves very high performances.

Weaknesses: If I understand correctly, the way to add "benign confounders" might introduce a slight bias to the dataset that, for each multimodal hateful memes, the image or text is likely to appear more than once in the dataset. For example, when a model sees an image which has been seen before, if the previous meme is non-hateful, this one is likely to be hateful; and vice versa, if an image is never seen before, it's more likely not hateful. I don't know if this will be a problem.

Correctness: The task is rigorously defined and the dataset collection is carefully designed and executed to ensure the quality of the dataset. The experiment design and results analysis are logically sound.

Clarity: The paper is clearly written and logically consistent.

Relation to Prior Work: The authors related this work to Hate Speech research, which is the practical field of this work; and Vision and Language tasks, which is the technical field. Another line of related work is the multimodal models. Many relevant models are mentioned in the Models section; however it may still be helpful to discuss it in Related Work to mention their progress and impact in practice.

Reproducibility: Yes

Additional Feedback: Typos: L236: (right bracket missing) ====== After author response ====== I have read the author response. I admit that the possible skew in dataset might not be an essential problem - I just wanted to raise this point. Indeed, it would be great if some analysis here could be added to the paper. I agree with other reviewers' ideas to add more analysis on the model performance on different type of samples, as well as a more detailed comparison with previous dataset (like [25]). These would be helpful too.

[Author Response · NeurIPS 2020]

We thank the reviewers for their thorough and very helpful feedback. We are glad that all reviewers found the dataset to be a valuable contribution—we believe that this work is important for providing better measurements for multimodal AI research in the future, with a clear positive contribution to society as a consequence. We address each reviewer below:

**Reviewer 1** Thank you for your insightful review, we will do our best to incorporate your excellent suggestions.

We will include a more detailed analysis of the dataset properties in the camera ready, if accepted, including of the dev set and a breakdown of multimodal vs unimodal hate, benign image/text, other random non-hateful. We did not do this initially because we wanted to avoid compromising our "unseen" dataset.

"An additional evaluation [..] using subsets of the training set of different sizes could shed some light" – Thank you for this excellent suggestion! We quickly did this experiment for the MMBT-Grid model and performance goes up considerably from using 10% of the training data (60.46 ROC-AUC on dev) to 50% (64.00) to 100% (68.57) of the training examples. We will include a plot in the camera ready, as well as provide further analysis.

We agree about real world meme generalization. Many such memes do use stock photos, however, and since we also release the raw SVG files it is easy to create different variations of the same meme, which is an interesting research direction. We will also add a column for easy/middle/late fusion to Table 1 to make that clearer.

The unimodal versions of VilBERT and Visual BERT are essentially the initializations used when pre/inter-training ViLBERT and VisualBERT models: rather than first training on multimodal data (e.g., COCO or Conceptual Captions), these models are finetuned directly on the Hateful Memes task without the intermediate training step.

**Reviewer 2** We really appreciate your thoughtful review and look forward to incorporating your comments.

We will include a plot of varying training dataset sizes in the camera ready, if accepted (see above). We will also include further analysis of the label quality as it relates to dataset size (our analysis for R1 above showed that even 10% of the training data is very useful, so you make a good point) – thanks for this suggestion. As you note, annotation was very costly, so this trade-off is definitely worth making explicit and examining further.

We agree that using images from a single source like Getty could make the distribution different from (some) real world memes. However, since the same procedure was used for all memes in the dataset, we think that it isn't a huge problem here, especially since many real memes are built using stock images as well. We also release the SVG files, so we hope that future work will try to analyze this further by replacing the background images and modifying the text properties.

An analysis of different model failure modes will be very interesting indeed—from what we have seen, the top models make similar mistakes, which will be useful to demonstrate in-depth, thanks for the suggestion.

Non-standard text is handled by the text-encoders: the transformer-based models all use Byte-Pair-Encoding, which means they are more robust to typographical errors, acronyms and out-of-vocabulary words, but you are definitely right that this would be a good avenue for trying to improve model performance on this task.

**Reviewer 3** Thank you for your review. We were a bit surprised by some of your points, which we hope to address:

Regarding the paper's organization: We respectfully disagree with your assessment—in fact the other reviewers all note that the paper is well written. We agree that this paper's contribution is different from more standard dataset papers (which we think is a good thing), which also means that we have to spend more time discussing the non-standard annotation process (i.e. in describing how we define hate speech or how we obtain benign confounders). We will happily include more dataset analysis, and will endeavor to make it even clearer what the dataset improves over previous work.

With regard to the binary label, we believe that this has several important benefits: i) it makes evaluation straightforward, which is important for machine learning problems, especially if we are trying to encourage the community to tackle an important problem together, for the greater good; and ii) as we describe in the paper, a binary label is actionable in practice: if a meme is hateful, it can be taken down; if a meme is disagreeable but ultimately not hateful, it should stay up – this distinction is ill-defined for an alternative finer-grained labelling. We agree that finer-grained labels can also be very valuable and should be investigated, but that question is unfortunately out of the scope of this work.

We respectfully disagree that the baseline models are too simple: we used state-of-the-art multimodal models, which are well-known as such in the V&L community. Note that MMBT, which uses grid features and is much simpler than ViLBert and VisualBert, compared to gated fusion in their paper and beat it; DeepDualMapper is specific to images and does not incorporate textual information. That said, we would happily include gated fusion as well in the camera ready.

**Reviewer 4** We thank you for your support and very useful feedback.

You are absolutely right that the benign confounders introduce a slight skew to the source images. Do note that the text will be different in each case, so if anything this skew makes the dataset even more difficult. You make an interesting point however, and we will examine if this has an impact in the camera ready, if accepted.

[Meta-Review · NeurIPS 2020]

I've read all reviews and author response and I would like to recommend acceptance. However, I would like to ask the authors to: 1) introduce some analysis, which most of the reviewers are asking for and was also promised in the author response 2) provide some comparison to the very recenlty introduced Gomez et al 2020 dataset. It is definately OK to have more datasets for the same topic (if any, the more the better), but I would say it's quite important to also have a feeling of what are there differences (minimally, on a high-level). R1/R2/R4 all think this is a very interesting contribution that can spur more work on detecting and dealing with hateful speech on social media. Moreover, reviewers agree that this dataset was constructed carefully so as to disadvantage uni-modal models, thus providing a good test bed for multimodal reasoning. R3 has expressed a number of concerns with regards to organization of the paper, use of simple baselines but also lack of analysis. While I disagree with the first 2 points, none of these points warrant rejection of the paper.